# An end-to-end framework for diagnosing COVID-19 pneumonia via Parallel Recursive MLP module and Bi-LTSM correlation

**Yiwen Liu**[1]                                                          YWLIU@MAIL.DHU.EDU.CN
[1] *College of Information Science and Technology, Donghua University, Shanghai, China*
**Wenyu Xing**[2]                                                      WYXING20@FUDAN.EDU.CN
[2] *School of Information Science and Technology, Fudan University, Shanghai, China*
**Mingbo Zhao**[*1]                                                      MZHAO4@DHU.EDU.CN
**Mingquan Lin**[†3]                                              MIL4012@MED.CORNELL.EDU
[3] *Weill Cornell Medicine, Cornell University, New York, USA*

**Editors:** Accepted for publication at MIDL 2023

## Abstract

To fully extract the feature information of lung parenchyma in Chest X-ray (CXR) images and realize the auxiliary diagnosis of Corona Virus Disease 2019 (COVID-19) pneumonia, this paper proposed an end-to-end deep learning model, which is mainly composed of object detection, depth feature generation, and multi-channel fusion classification. Firstly, the convolutional neural network (CNN) and region proposal network (RPN)-based object detection module was adopted to detect chest cavity region of interest (ROI). Then, according to the obtained coordinate information of ROI and the convolution feature map of original image, the new convolution feature maps of ROI were obtained with number of 13. By screening 4 representative feature maps form 4 convolution layers with different receptive fields and combining with original ROI image, the 5-dimensional (5D) feature maps were generated as the multi-channel input of classification module. Moreover, in each channel of classification module, three pyramidal recursive multilayer perceptrons (RMLPs) were employed to achieve cross-scale and cross-channel feature analysis. Finally, the correlation analysis of multi-channel output was realized by bi-directional long short memory (Bi-LSTM) module, and the auxiliary diagnosis of pneumonia disease was realized through fully connected layer and SoftMax function. Experimental results show that the proposed model has better classification performance and diagnosis effect than previous methods, with great clinical application potential.

**Keywords:** Pneumonia, end-to-end framework, multi-task learning, RMLP-Bi-LSTM, deep learning.

## 1. Introduction

Corona Virus Disease 2019 (COVID-19) was discovered at the end of 2019 and began to break out in a large scale in 2020. So far, it is still a powerful and threatening infectious disease that plagues the lives and health of people all over the world (Frid-Adar et al., 2021). At present, the detection of COVID-19 diagnosis mainly depends on antigen and nucleic acid detection(Ji et al., 2020). They all have some limitation, such as high false positive or

---

[*] Corresponding author

[†] Corresponding author

low efficiency. Due to the timeliness and visibility of medical imaging, the combination with artificial intelligence has become an important means for detecting COVID-19 in clinic.

With the advantages of fast, low radiation, and high resolution, Chest X-ray (CXR) has been widely used in the diagnosis of patients with COVID-19 and other pneumonia. At present, there are many researches based on CXR images and artificial intelligence, and most of them used deep learning as the main diagnosis framework. For example, Monday et al. (Monday et al., 2022) proposed a super-resolution-based Siamese wavelet multiresolution convolutional neural network (CNN) model for COVID-19 diagnosis. Fan et al. (Fan et al., 2021) designed a multi-kernel-size spatial-channel attention deep learning model to detect COVID-19 pneumonia. Nikolaou et al. (Nikolaou et al., 2021) proposed a new CNN framework by adding a dense layer to the pre-training baseline CNN (EfficientNet-B0) to achieve high-precision classification of COVID-19. Priyatharshini et al. (Priyatharshini et al., 2021) used U-Net to segment the CXR image, and then used Inception v3 model to train the classification model to distinguish COVID–19. Li et al. (Li et al., 2021)proposed a capsule network model with multi-head attention routing algorithm (MHA-CoroCapsule) for accurate diagnostics of COVID-19. Agarwal et al. (Agarwal et al., 2021) proposed a two-stage deep learning diagnosis framework, CoroNet, to automatically perform the diagnosis task of COVID-19. Park et al. (Park et al., 2021) proposed a multi-scale class residual attention (MCRA) network based on CXR images, which can effectively detect COVID-19.

Although above researches have achieved good effect in the diagnosis of COVID-19, there are still some limitations that limit their clinical application. For example, for those end-to-end diagnostic models composed of a single depth learning module, the CXR images are directly processed, which cannot avoid the influence of other tissues in chest region. Moreover, the feature analysis using convolution alone cannot achieve a comprehensive analysis, and the extracted features have not been filtered, so there is some redundancy for subsequent feature analysis and calculation.

Therefore, in order to address those problems, this paper proposed an end-to-end deep learning model based on the combination of convolution, multilayer perceptron (MLP) and bidirectional long short memory (Bi-LSTM) modules to achieve the auxiliary diagnosis of COVID-19 and other pneumonia. The main contributions are summarized as follows.

- The end-to-end model can achieve the tasks of object detection, feature extraction, classification and recognition synchronously. Compared with the multi-stage cascade model, it has better operational efficiency and can better meet the clinical needs.

- By embedding Region Proposal Network (RPN)-based object detection module in the end-to-end model, it realizes the precise positioning of the chest region and reduces the interference of other regions and the calculation of subsequent features.

- The convolution module with different receptive fields and the multi-scale recursive MLP (RMLP) module achieve more sufficient local and global information extraction of CXR images. Bi-LSTM also increases the correlation of the output of different channel, improving the accuracy of auxiliary diagnosis.

## 2. Methodology

### 2.1. Overall framework

This paper proposed a novel end-to-end classification model for diagnozing COVID-19 and other pneumonia, which is mainly composed of a multilayer CNN-based object detection and feature map generation module and RMLP-Bi-LSTM-based classification module, as shown in Figure 1.

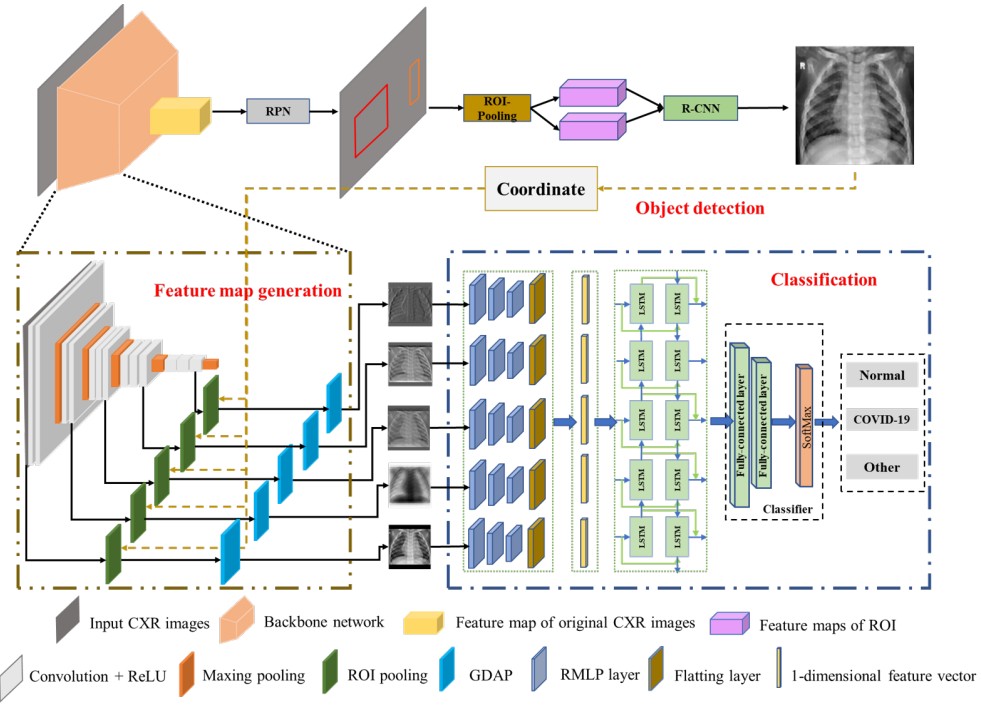

Figure 1: Schematic diagram of end-to-end classification framework.

### 2.2. Multi-task learning module for object detection and feature map generation

Due to the influence of patients' differences, operating habits, and types of instruments, the quality of CXR obtained from patients is different, and the proportion of chest region of interest (ROI) in CXR are different. In order to reduce the impact of these factors, follow-up research needs to be carried out on the basis of accurate positioning of the thoracic cavity. Therefore, this paper designed a multi-task learning module that integrates object detection and feature genetation, as shown in the object detection part and feature map generation part of Figure 1.

For object detection of thoracic cavity ROI, feature map of original image was firstly generated through CNN backbone module, which was put into RPN to obtain the proposal box of object area. Then, according to the combination of proposal box and feature map, ROI pooling was employed to re-extract a fixed size feature map for each proposal box.

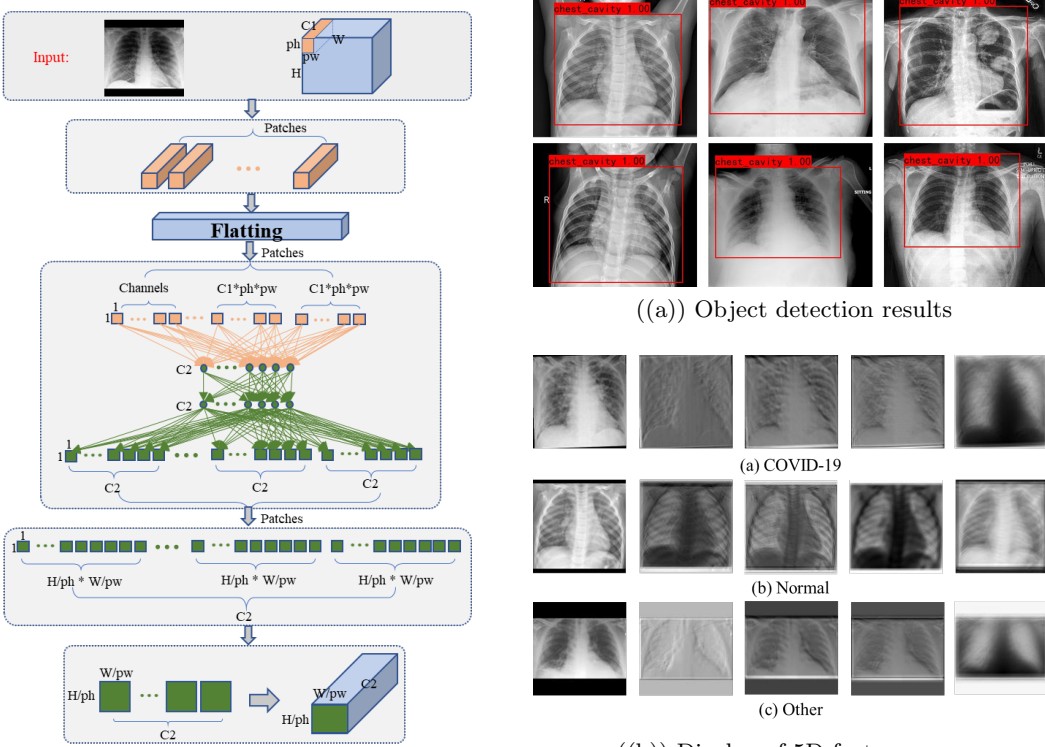

((a)) Object detection results

(a) COVID-19

(b) Normal

(c) Other

((b)) Display of 5D feature maps

Figure 2: Schematic diagram of the structure of the RMLP layer.

Figure 3: Display of Object detection and feature map generation.

Next, according to these feature maps, the objects in the proposed box were classified (i.e., background or chest) through fully connected layer and SoftMax function. Finally, the boundary information of ROI was further adjusted according to the predicted category to obtain accurate coordinate information.

For feature map generation, according to the obtained coordinate information and the convolution feature map of original image, the new feature maps of ROI were obtained. The CNN backbone model used in this paper contains 13 convolution layers. The convolution layers of four different receptive fields was screened for selecting the feature map of the last layer of each receptive field, where a Global Depth Average Pooling (GDAP) layer was designed to change the feature map with the size of $(c, h, w)$ to a single channel feature map with the size of $(h, w)$. Finally, the ROI feature maps of four convolution layer with different receptive field were combined with original ROI image to generate a new 5-dimensional (5D) feature input for RMLP-Bi-LSTM classification model.

## 2.3. RMLP-Bi-LSTM classification module

MLP-Mixer was proposed by Google Brain Team in 2021 (Tolstikhin et al., 2021) which is fully based on MLP. It has been applied in many fields, which put the token of input images into token-mixer layer and channel-mixer layer in turn to mix the information of both

feature channels and spatial locations. Inspired by this, we proposed a pyramid RMLP-Bi-LSTM classification module with multi-channel fusion under multidimensional input to achieve cross-channel and cross-space feature analysis and classification of input, as shown in the classification part of Figure 1. The 5D feature input generated by the multi-task module was input into three serial RMLP layers with different scales in parallel to analyze the channel and spatial information of the input. Then the output of the RMLP layers was flattened, and five parallel outputs were spliced to generate sequence information, which is transmitted to Bi-LSTM to explore the correlation between its channels. Finally, the classification detection of COVID-19 patients was completed through two full connection layers and SoftMax layer.

The structure of the RMLP layer is shown in Figure 2. Assuming the size of the input feature is $(b, C1, H, W)$ in each RMLP layer. Firstly, the dimension of the feature map was reduced. The input was divided into patches with the size of $(ph, pw)$, and then the patches were expanded into one-dimensional vectors The input was converted from $(b, C1, H, W)$ to $(b, H/ph \times W/pw, ph \times pw \times C1)$ to realize the dimension reduction operation. Then the feature extraction of one-dimensional vector was realized through the MLP layer composed of two layers of fully connected layers with $C2$ neurons. Finally, through the dimension elevation layer, we completed the spatial information reorganization of the feature by restoring $(b, H/ph \times W/pw, C2)$ to the data form consistent with the original input dimension $(b, C2, H/ph, W/pw)$, as the input of the next RMLP layer. Here, $ph = pw = 2$, and the input patch sizes of the three RMLP layers are $n1 \times n1$, $n2 \times n2$, $n3 \times n3$ respectively, where $n3 = 2 \times n2 = 4 \times n1$. Assume the size of input feature map was $M \times M \times 1$, the output sizes of three pyramidal RMLPs are $M/2 \times M/2 \times 16$, $M/4 \times M/4 \times 32$, and $M/8 \times M/8 \times 64$.

## 3. Experiments and Results

Experimental dataset used in this paper includes 1341 cases of normal, 183 cases of COVID-19 patients, and 1294 cases of other pneumonia patients (Minaee et al., 2020; Tahir et al., 2021). To balance the number of samples in different categories, the images of COVID-19 cases were expanded to 1464 by random angle rotation. As a result, a total of 4099 images (Normal: 32.71%, COVID-19: 35.72%, Other: 31.57%) were used in this experiment. Meanwhile, 10-fold cross validation method was used in experiments, which was repeated 5 times, with the mean value as final results.

### 3.1. Object detection and feature map generation

Through the multi-task learning module combining the pre-trained 13-layer convolutional neural network and RPN, the 13-dimensional (13D) depth feature map of thoracic cavity ROI can be generated. Meanwhile, in order to reduce the computational complexity, the characteristics of four different receptive fields were screened in experiments. Then through the GDAP layer, four fixed-size outputs of different depths were obtained, which were combined with original ROI image to form a 5D feature input, as shown in Figure 3.

### 3.2. Qualitative Results of classification experiments

The above 5D features were input into the multi-channel RMLP-Bi-LSTM classification module, with training hyperparameter of epoch, learning rate, batch size, and optimizer being 30, 0.0005, 32, and AdamW, respectively. The data encoding method adapt one-hot encoding for calculating cross entropy loss function value to obtain the classification results for normal, COVID-19 and other pneumonia.

After five repeated experiments, mean value of five indicators (i.e., Accuracy, Precision, Specificity, Recall, and F1-socre) were obtained to quantitatively evaluate the classification performance. Experimental results are shown in Table 1, which show that the proposed model has superior classification performance, with five indicators being 98.10±0.68, 98.13±0.67, 99.03±0.35, 98.30±0.83, and 98.12±0.67, respectively.

Table 1: Quantitative evaluation results of the proposed model.

| Order | Indicator value (%) | | | | |
|---|---|---|---|---|---|
| | Accuracy | Precision | Specificity | Recall | F1-score |
| 1 | 98.54 | 98.57 | 99.23 | 99.35 | 98.61 |
| 2 | 97.81 | 97.69 | 98.88 | 97.90 | 97.75 |
| 3 | 97.81 | 98.00 | 98.91 | 97.85 | 97.89 |
| 4 | 99.03 | 99.01 | 99.52 | 98.99 | 99.00 |
| 5 | 97.32 | 97.36 | 98.61 | 97.40 | 97.36 |
| Avreage±STD | 98.10±0.68 | 98.13±0.67 | 99.03±0.35 | 98.30±0.38 | 98.12±0.67 |

### 3.3. Analysis of model ablation experiment

#### 3.3.1. Feasibility of 5D feature selection strategy

In order to verify the feasibility of 5D input composed of the 4-dimensional (4D) feature map selected from four convolutional layers with different receptive fields and original image, we compared 5D input with original 14-dimensional (14D) input (Figure 4), which was composed 13D feature map output from CNN backbone and original image. The performance and comparison of the classification model trained through 14D and our proposed model was shown in Table 2, with gain of 1.02%, 1.03%, 0.48%, 1.29%, and 1.09% respectively in accuracy, precision, specificity, recall, and F1-score, which proved that the 5D feature used in the proposed model can replace the 14D feature, which can not only reduce calculation complexity, but also improve the classification accuracy.

#### 3.3.2. Effectivity of Bi-LSTM module

In the classification model, a feature extraction module based on RMLP was designed to carry out more in-depth feature analysis of multi-channel input images, and Bi-LSTM was adopted to enhance the correlation between different input channels. In order to validate the effectivity of Bi-LSTM module, this paper conducted the same experiment using 5D input on the classification model without Bi-LSTM module. The quantitative evaluation results of the model of 5D feature inputs without Bi-LSTM and comparison with our proposed model are shown in Table 3, with gain of 3.80%, 3.66%, 1.91%, 4.08%, and 3.85% respectively

in accuracy, precision, specificity, recall, and F1-score, which proved that the Bi-LSTM module can capture the correlation between different input channels and greatly improves the model's classification performance.

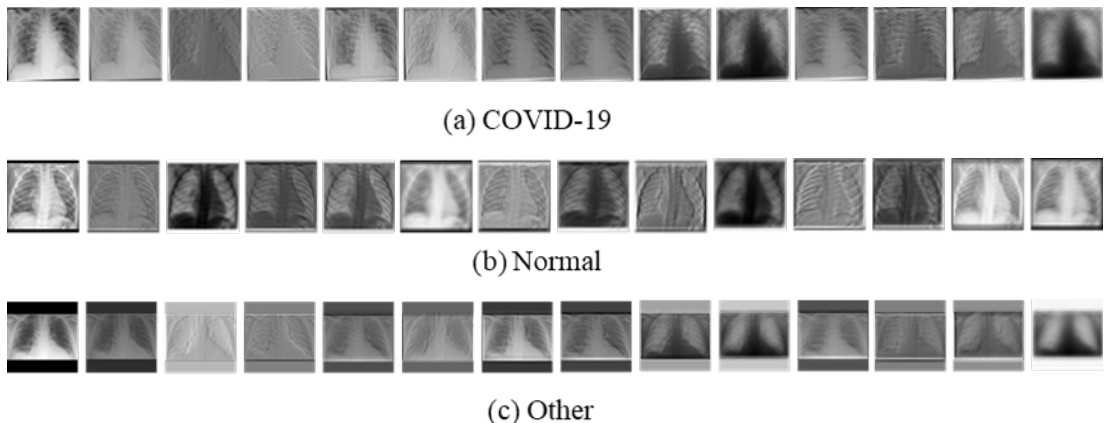

Figure 4: Display of 14D feature maps.

### 3.4. Comparison with other deep learning models

In order to verify the superiority of the method proposed in this paper, it was compared with previous classical deep learning models, such as Visual attention network (VAN) (Guo et al., 2022), ConvNet (Liu et al., 2022), MobileNet-V3 (Howard et al., 2019), DenseNet (Huang et al., 2017), MLP-Mixer (Tolstikhin et al., 2021), and Vision Transformer (ViT) (Dosovitskiy et al., 2020). Experimental results were shown in Table 4, which proved that the proposed model has significantly improved performance and can more accurately realize the auxiliary diagnosis of COVID-19, with great clinical application potential.

## 4. Conclusion

In this paper, we proposed a novel end-to-end deep learning framework to achieve rapid and accurate diagnosis of COVID-19. This framework realized the integration of object detection and feature generation, which can effectively eliminate the interference of other regions and generate multi-source feature maps. Meanwhile, in the pyramid RMLP-Bi-LSTM classification module with multi-channel input, not only the deep abstract feature extraction of the multi-channel input was completed, but also the correlation between multi-channel input was combined, which further enhanced the information analysis of proposed model on CXR image, and realized the high-precision classification diagnosis of COVID-19, with great clinical application potential.

## Acknowledgments

This work is supported by the National Natural Science Foundation of China (61971121).

Table 2: Quantitative evaluation results of the 14D feature inputs.

| Order | Indicator value (%) | | | | |
|---|---|---|---|---|---|
| | Accuracy | Precision | Specificity | Recall | F1-score |
| 1 | 96.11 | 96.31 | 98.13 | 95.69 | 95.96 |
| 2 | 98.54 | 98.53 | 99.27 | 98.54 | 98.53 |
| 3 | 96.11 | 96.03 | 98.02 | 96.09 | 96.08 |
| 4 | 97.32 | 97.28 | 98.64 | 97.39 | 97.31 |
| 5 | 97.32 | 97.34 | 98.67 | 97.33 | 97.27 |
| 14D | 97.08±1.02 | 97.10±0.99 | 98.55±0.50 | 97.01±1.14 | 97.03±1.05 |
| Ours | 98.10±0.68 | 98.13±0.67 | 99.03±0.35 | 98.30±0.38 | 98.12±0.67 |

Table 3: Quantitative evaluation results of the 5D feature inputs without Bi-LSTM.

| Order | Indicator value (%) | | | | |
|---|---|---|---|---|---|
| | Accuracy | Precision | Specificity | Recall | F1-score |
| 1 | 95.13 | 95.48 | 97.59 | 95.18 | 95.18 |
| 2 | 94.89 | 94.96 | 97.36 | 94.98 | 94.93 |
| 3 | 92.94 | 93.02 | 96.34 | 92.94 | 92.92 |
| 4 | 93.43 | 93.85 | 96.78 | 92.91 | 93.24 |
| 5 | 95.13 | 95.06 | 97.55 | 95.10 | 95.07 |
| Without Bi-LSTM | 94.30±1.04 | 94.47±1.01 | 97.12±0.54 | 94.22±1.19 | 94.27±1.09 |
| Ours | 98.10±0.68 | 98.13±0.67 | 99.03±0.35 | 98.30±0.38 | 98.12±0.67 |

Table 4: Comparative results of various models.

| Model | Indicator value (%) | | | | |
|---|---|---|---|---|---|
| | Accuracy | Precision | Specificity | Recall | F1-score |
| VAN | 76.28 | 78.84 | 88.83 | 77.20 | 75.71 |
| ConvNet | 78.00 | 78.65 | 89.20 | 78.67 | 77.52 |
| MobileNet-V3 | 89.24 | 89.91 | 94.51 | 89.32 | 89.46 |
| DenseNet | 85.21 | 86.5 | 92.74 | 85.66 | 85.24 |
| MLP-Mixer | 95.23 | 95.34 | 97.64 | 95.50 | 95.25 |
| ViT | 90.46 | 91.35 | 95.53 | 91.02 | 90.37 |
| Ours | 98.10 | 98.13 | 99.03 | 98.30 | 98.12 |

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
