# OpenReview forum: "An end-to-end framework for diagnosing COVID-19 pneumonia via Parallel Recursive MLP module and Bi-LTSM correlation"
_MIDL.io/2023/Conference — MIDL 2023 Oral_

### Official Review · Reviewer_RqiG · 2023-02-03

**Confidence:** 4
**Preliminary Rating:** 2

**Summary:**

This paper proposes a new model for diagnosing COVID-19 pneumonia from chest X-ray images. The model consists of three main parts: object detection, feature generation, and multi-channel classification.
Object detection through RPN intends to isolate the chest cavity, then the feature generation section creates 5-dimensional features from such region. Each feature map inputs into a three layers MLP pyramid which intends to extract multi-channel and multi-scale features that are transformed into a new set of compact features based on the correlations given by a Bi-LSTM module. Finally, the features are employed to classify the image as normal, COVID-19 or other pathology.

**Strengths:**

- The model follows a nice engineering approach combining different usual "pieces" in state-of-the-art processing pipelines. This requires a great effort of implementation and adaptation to the resources usually available.
- The model stages are clear which facilitates a similar implementation.
- The results are promising and therefore make it worth trying the model in new datasets.
- The same approach could be easily adapted for other CXR task-related.

**Weaknesses:**

- Overall, the text seems to be written hastily and can be confusing at times. For example, the authors have not filled in the "Short Title" field.
-  The explanation of the stages of the model is simple to understand, but the important part, the implementation, is missing. There are also missing details about the model's design, which makes it hard to recreate even if the stages are clear.
- A big concern for me is the lack of validation, especially in a such problem as COVID detection. There are numerous reputable datasets available to the public$^{1,2}$.  The authors' choice to only analyze a single dataset with such a limited number of real positive cases raises questions.

- If object detection proves to be useful, could it be due to the well-known problem of spurious correlations$^3$ in this type of dataset instead of the influence of other tissues as the authors claim?
- Authors claim multi-tasking but just one task is coped disease classification.




[1] Costa et al. "COVID-19 Detection on Chest X-ray and CT Scan: A Review of the Top-100 Most Cited Papers". Sensors. 2022

[2] Kaggle. "COVID-19 Radiography Database". 2021


[3] De grave et al. "AI for radiographic COVID-19 detection selects shortcuts over signal". Nature Machine intelligence. 2021

**Deanonymize Review:**

yes

**Detailed Comments:**

Given the number of reviews and the sort of rebuttal period, I prefer to avoid minor and detailed comments.  I hope that both the authors and the AC are in agreement with this. If you need clarification on any point, please let me know.

**Paper Type:**

methodological development

**Questions To Address In The Rebuttal:**

  - Please upload the code to an anonymous repository for ease of access.
- Could the authors provide at least a docker for inference?
- The state of surrounding tissues could be informative in medical imaging tasks, however, such information is eliminated by proposing an RPN, have you performed an ablation study eliminating the object detection stage in your model?
- Could a multi-task loss (object detection loss + Cross Entropy loss) improve the results?
- Where in the architecture does the author incorporate a mechanism to consider the claimed cross-scale effect?
- For clarity, Figures 1 and 2 should be separated into smaller figures and referred to in the text when different stages are discussed.
- Probably you can save figure three and orientate vertically the figure 2.
- With all due respect, the title of section 3.2 is almost robotic.
- Dispersion measures are given as the standard deviation on each fold repetition average. This could be misleading and it would be helpful to at least include the dispersion per fold and repetition.

---

### Official Review · Reviewer_1SMW · 2023-02-04

**Confidence:** 3
**Preliminary Rating:** 5
**Recommendation:** Best Paper Award

**Summary:**

To fully extract the feature information of lung parenchyma in Chest X-ray images and realize the auxiliary diagnosis of Covid19 pneumonia, this paper proposed an end-to-end deep learning model, mainly composed of object detection, depth feature generation, and multi-channel fusion classification. Experimental results show that the proposed model has better classification performance and diagnosis effect than previous methods, with great clinical application potential.



**Strengths:**

-	The proposed end-to-end deep learning model can achieve the tasks of object detection, feature extraction, classification, and recognition synchronously.
-	By embedding an RPN-based object detection module in the end-to-end model, it realizes the precise positioning of the chest region. It reduces the interference of other regions and the calculation of subsequent features.
-	The convolution module with different receptive fields and the multi-scale MLP module achieve local and global information extraction.
-	High-precision classification


**Weaknesses:**

-	The Covid19 images have been augmented just by random rotation.
- The text in the figures is too little to be readable.
- The graph in figure 2 is rotated 90 degrees, making it difficult to visualize it.


**Deanonymize Review:**

no

**Paper Type:**

both

**Questions To Address In The Rebuttal:**


-	Please, define all the acronyms the first time they appear in the text.
-	The input dataset was heavily unbalanced (1341 control, 183 Covid19, 1294 other pneumonia). Why were rotations the method used for data augmentation? Why not scale changes or noise addition?

---

### Official Review · Reviewer_uA9F · 2023-02-05

**Confidence:** 5
**Preliminary Rating:** 3

**Summary:**

This paper achieves the tasks of COVID-19 pneumonia detection, feature extraction, classification and recognition synchronously. Experimental results show that the proposed model has better classification performance and diagnosis effect than previous methods, with great clinical application potential. On the whole, putting multi-task on COVID19 is strong for the problem, but the motivation of the structural design is insufficient, such as splicing some deep learning models.

**Strengths:**

1) This framework mainly composed of object detection, depth feature generation, and multi-channel fusion classification, which plays a role in clinical application potential.
2) Experimental results show that the proposed model has better classification performance and diagnosis effect than previous methods

**Weaknesses:**

1) The title and the structure of framework feels like connecting multiple deep learning models in series, making it difficult to grasp the focus and contribution.
2) The motivation of introducing the MLP-Mixer is insufficient.
3) Operational efficiency compared to multi-stage cascaded models is not supported by experiments

**Deanonymize Review:**

no

**Detailed Comments:**

1) Is there anything special about generating 5-dimensional (5D) feature maps, and what about generating feature maps of larger or smaller dimensions?
2) More motivations of introducing the MLP-Mixer need to be explained.

**Paper Type:**

methodological development

**Questions To Address In The Rebuttal:**

1) The title seems not match the attraction of this paper and cannot highlight the contributions.
2) Is there anything special about generating 5-dimensional (5D) feature maps, and what about generating feature maps of larger or smaller dimensions?
3) More motivations of introducing the MLP-Mixer need to be explained.
4) The source and presentation of the data requires explanation.
5) How does it compare to the multi-stage cascade model in terms of parameter size and execution efficiency?

---

### Meta-Review · Area_Chair_cNxv · 2023-02-24

**Recommendation:** Accept (Poster)
**Confidence:** 5

**Metareview:**

Overall, all reviewers are satisfied with the response given by the authors, and are glad to see that the quality of the paper has been improved substantially.